# Caregiver perceived barriers to the use of micronutrient powder for children aged 6–59 months in Bangladesh

**Mahfuzur Rahman** [1]*, **Md. Tariqujjaman**[1], **Mustafa Mahfuz**[1], **Tahmeed Ahmed** [1], **Haribondhu Sarma** [2]

**1** Nutrition and Clinical Services Division, icddr,b, Mohakhali, Dhaka, Bangladesh, **2** Research School of Population Health, Australian National University, Canberra, Australian Capital Territory, Australia

* mahfuzur.rahman@icddrb.org

**Data Availability Statement:** Data generated from icddr,b's research can be provided to interested researchers (Recipients) for secondary data analyses upon approval of a Data Licensing

## Abstract

### Background

The effectiveness of micronutrient powder (MNP) on the health outcome of children is yet to be proved. Although studies identified the barriers to the use of MNP the underlying factors related to the barriers to the use of MNP are still unexplored. We examined the underlying factors associated with the barriers reported by the caregivers of the children aged 6–59 months in Bangladesh.

### Methods

We analyzed pooled data of 3, 634 caregiver-child dyads extracted from eight cross-sectional surveys. The surveys were conducted as part of an evaluation of the Maternal, Infant and Young Children Nutrition programme (phase 2) in Bangladesh. We performed univariate analysis to find the barriers reported by the caregivers of the children. We identified the underlying factors related to the reported barriers by performing multiple logistic regression analysis.

### Results

The mostly reported barrier was perceived lack of need for MNP among the caregivers of the children (39.9%), followed by lack of awareness of the product (21.7%) and cost of the product (18.1%). Caregivers of older children (adjusted odds ratio (aOR): 1.69; 95% CI: 1.43, 2.00) and caregivers who maintained good infant and young child feeding practices (aOR: 1.32; 95% CI: 1.12, 1.57) perceived more lack of need for MNP. Caregivers of the female children (aOR: 0.79; 95% CI: 0.63, 0.98) were less likely to report that their children disliked MNP compared to the caregivers of the male children.

### Conclusion

Programmes intended to effectively promote MNP among the caregivers of children aged 6–59 months should carefully consider the factors that could underlie the barriers to the use of MNP.

Application & Agreement by the icddr,b Data Centre Committee. Interested personnel is recommended to consult this with icddr,b IRB Coordinator Mr. M A Salam Khan (salamk@icddrb. org).

**Funding:** Research for this article was founded by the Children's Investment Fund Foundation (CIFF), UK. The views, opinions, assumptions, or any other information set out in this article are solely those of the authors and should not be attributed to CIFF or any person connected with CIFF.

**Competing interests:** The authors have declared that no competing interests exist.

## Introduction

Globally, more than one and half billion people are anaemic and most of them are pre-school aged children and pregnant women [1]. Pre-school aged children are at risk of anaemia due to their rapid growth and low consumption of iron and micronutrient dense foods [2]. Children need adequate vitamin and mineral for their development at least up to two years of age [3]. Due to suboptimal practices of infant and young child feeding (IYCF) and less diversity in dietary practices, it fails to meet the recommended nutrient intakes that may lead to immediate and long-term consequences [4]. Moreover, plant-sources food cannot provide sufficient micronutrients, particularly iron. On the other hand, the inclusion of animal-source foods in dietary practice, to reduce micronutrient deficiency, may not be affordable for the lower income groups [5, 6]. Thus, it necessitates introducing multiple micronutrient powder (MNP) for improving nutritional status and reducing anaemia.

Although the efficacy of MNP in reducing childhood anaemia has already been established, however, the effectiveness of MNP usage at the community level is yet to be proven [7]. Programmes promoting MNP confronts barriers from the perspective of both supply and demand side, and their effectiveness rely upon the delivery channels or programme models they use [8–10]. One of the potential delivery channels is a market-based approach to promoting MNP through volunteer community health workers [8] and this approach has been found to be cost-effective [11]. However, the study indicates that whatever the delivery channel is, uninterrupted supplies and programmatic inputs and training of the providers are in central to make the MNP promotional programme effective [8]. Lack of programmatic inputs and insufficient training of the provider- particularly of community health worker (CHW) can result in low uptake of MNP at the community level [12]. The study also has shown that individual, community and organizational level factors may pose barriers for the CHWs to perform in promoting MNP including a lack of self-efficacy, family support, and the provision of capacity building for income generation [13]. These aforementioned supply side barriers to coverage of MNP are evident in low- and middle-income countries [10]. However, none of the studies did focus on how these factors may lead to creating barriers for the caregivers of the children who are the ultimate beneficiaries of MNP.

Apart from these supply side barriers, studies revealed barriers to coverage of MNP and other nutrient supplement from the perspective of demand side [14, 15]. These demand side barriers are related to the perception of the caregivers of the children and their socio-cultural factors. Although several studies have been done to identify the barriers to use of MNP, to our knowledge, however, no study has been conducted so far to identify the underlying factors relating to the barriers to the use of MNP among the caregivers of the children aged 6–59 months. It is imperative to identify the factors contributing to the barriers to the use of MNP so that interventions could be devised for overcoming the barriers and thus MNP promotional programme could be made more effective. In this article, we present the barriers to the use of MNP reported by the caregivers of the children under a community-based Maternal Infant and Young Child Nutrition (MIYCN) Phase-2 Programme in Bangladesh and the underlying factors related to those barriers.

Bangladesh MIYCN programme was a collaborative endeavor of GAIN (Global Alliance for Improved Nutrition) and BRAC-a non-governmental organization based in Bangladesh. The programme components included enabling policy environment for home fortification with Pushtikona-5 (a brand name of MNP), improving the delivery channels to ensure effective coverage of Pushtikona-5 through CHW of BRAC and demand creation to scale up home fortification practices and demand for Pushtikona-5 [16]. Key interventions for improving the delivery channel for MIYCN in order to ensure effective coverage of Pushtikona-5 through

community health workers (CHWs) of BRAC included training (a basic training workshop and monthly refresher training) to CHWs to promote home-fortification of diets with Pushti-kona-5 at the household level, counseling to the caregivers of children aged 6–59 months, on home-fortification with Pushtikona-5, monitoring home-fortification activities at the community level by CHW and stimulation CHWs with the provision of incentives to promote Pushti-kona-5 [16]. Under the MIYCN programme, BRAC sells Pushtikona-5 to a CHW at US$ 0.02 per sachet and asks the CHW to sell it to the caregivers at US$ 0.03 per sachet. The programme aimed to reduce the prevalence of anaemia among under-5 children by 10% by the end of the programme.

icddr,b was the evaluation partner in the programme. Results presented in this paper are part of a large evaluation of the MIYCN programme, undertaken by icddr,b. The comprehensive account of the large evaluation of the MIYCN programme has been published elsewhere [17]. We anticipate that the findings described in this paper will provide directives for the implementers to design an effective MNP promotion programme.

## Materials and methods

### Study design and setting

Eight cross-sectional surveys were carried out from 2014 to 2018 as part of an evaluation of Bangladesh MIYCN programme. Surveys were conducted at the household level among the caregivers of the children of 6–59 months in 26 administrative districts of Bangladesh where BRAC's MIYCN programme was implemented.

### Study population

Caregivers of the children of 6–23 months were the study population in this study. A caregiver was defined as the child's biological mother or the person who took care or looked after and gave the child most meals on most days. The inclusion criteria of selecting the household were the households having at least a child of 6–59 months. If a household had more than one child of 6–59 months a child was selected in a random selection process by lottery. The survey excluded the households if the caregiver or child belonged to the households was physically or mentally challenged, or the child had any disease during the day of the survey.

### Data collection and extraction

A two-stage sampling procedure was applied to select a total of 16, 936 caregiver-child dyads and data were collected using Open Data Kit. The details of the sampling and data collection process are mentioned in elsewhere [18]. For this article, we analyzed data of 3, 634 samples who responded that they had seen or heard of Pushtikona-5, but did not feed to their children. Data extraction process is presented in "Fig 1".

### Variables measured

Household level information such as the number of the household member and the assets in the household was obtained during the survey. Households were categorized into poor, middle and rich based on the wealth index. We calculated the wealth index based on the household's ownership of selected assets, household structure (materials used for floor, roof, and wall of the house), type of latrine used and sources of drinking water by using principal component analysis [19].

Child's age was calculated from his or her date of birth mentioned in the health card or reported by the caregiver when the health card was unavailable. Caregivers' age and age of the

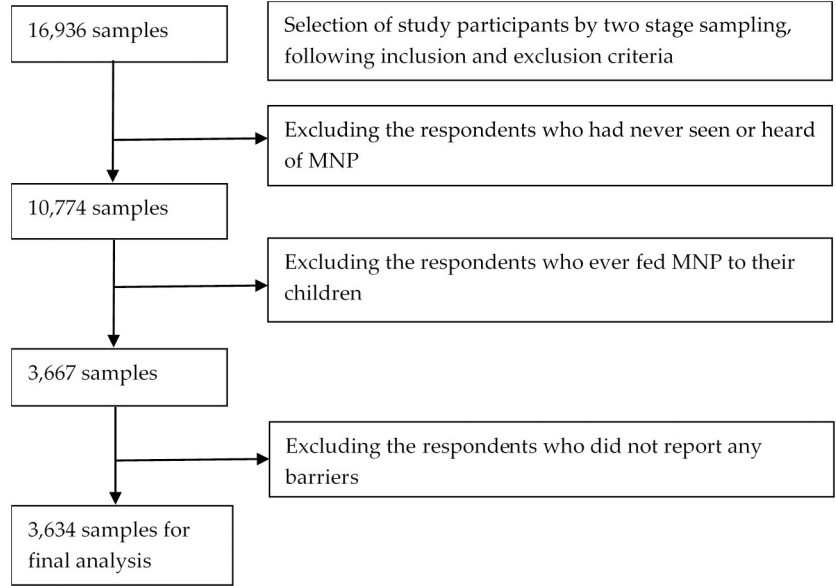

**Fig 1. Extraction of samples for analysis.**

child's father were calculated from the date of birth mentioned in their national identity cards. Child age was categorized as 6–23 months and 24–59 months. Caregivers' age was categorized as <25 years and ≥ 25 years. Child's father age was categorized as <30 years and ≥30 years. Father's and caregiver's age were categorized based on the median value.

Caregivers were asked if they had ever seen or heard of MNP (any brand including Pushto-kona-5) and if they ever fed MNP to their children prior to the day of survey. Respondents, who had seen or heard of MNP but never fed to their children, were asked why they did not feed. The reasons they mentioned for not feeding MNP to their children were considered as their barriers to the use of MNP. Multiple barriers were reported by the caregivers. A 24-hour dietary recall questionnaire was used to assess Infant and Young Child Feeding (IYCF) practices. IYCF practices were scored based on the Infant and Child Feeding Index (ICFI) [20]. If the ICFI score was maximum (ICFI score = 6) the IYCF practices were considered as optimal "S1 Table". We also asked the caregivers if their households were visited by a CHW in the last 12 months prior to the day of the interview.

## Data analysis

We performed univariate analysis and tabulated the frequency and proportion of different variables. Multivariable logistic regression analysis was performed to find the factors associated with different barriers to the use of MNP reported by the caregivers and odd ratios along with 95% confidence interval were tabulated. The potential variables that were found significant at p-value < 0.20 in the simple logistic regression model were kept in the multivariable logistic regression model. Data were analyzed using STATA version 13.0 (Stata Corp, 4905 Lakeway Drive, College Station, Texas 77845, USA).

## Ethics statement

Study protocol was reviewed and approved by the Institutional Review Board of icddr,b which is comprises of Research Review Committee and Ethical Review Committee. Well-informed

written consents were obtained from the caregivers of the children before starting the survey. The Ethical Review Committee approved the consent form prior to the survey.

## Results

A total of 3634 (21.5%) caregivers mentioned that they had ever seen or heard of MNP but did not feed MNP to their children. Their socio-demographic characteristics are presented in "Table 1". Of their households, 58.9% had ≥ 5 members. Among the children of 6–59 months, 51.7% were male. Two or more children of 6–59 months were found in 17.5% of households. The remaining 82.5% of households had one child of 6–59 months. Caregivers and fathers of the children who completed ≥5 years of schooling were 78.7% and 64.4% respectively.

"Fig 2" shows the barriers to the use of MNP reported by the caregivers. Barriers reported by most of the caregivers were perceived lack of need (39.9) followed by lack of awareness of the product (21.7%), cost of the product or not affordable (18.1%), irregular or insufficient supply of MNP (13.1%) and disliking the product by the children (12.7%). However, discouraged by the neighbors or family members (4.3%) also came up as a barrier reported by the caregivers.

"Table 2" presents the factors associated with the barriers to the use of MNP among the caregivers. Caregivers of the older children (aOR: 1.69, 95% CI: 1.43, 2.00), caregivers of Muslim families (aOR: 1.43, 95% CI: 1.05, 1.95) and caregivers of the middle (aOR: 1.45, 95% CI: 1.19, 1.77) and the rich (aOR: 1.81, 95% CI: 1.46, 2.23) households were more likely to perceive lack of need for MNP for their children. Caregivers who maintained optimal IYCF practices were 32% more likely to perceive lack of need for MNP for their children compared to the caregivers who did not maintain optimal IYCF practices. Results showed, although

**Table 1. Socio-demographic characteristics of the study participants (n = 3634).**

| Variables | Frequency | Percentage |
|---|---|---|
| Household size | | |
| ≥ 5 members | 2141 | 58.9 |
| No. of children (6–59 months) in the households | | |
| Two or more | 637 | 17.5 |
| Sex of the children | | |
| Male | 1860 | 51.7 |
| Child's age | | |
| 24–59 months | 1888 | 52.8 |
| Caregiver's religion | | |
| Muslim | 3292 | 90.3 |
| Caregiver's education | | |
| ≥ 5 years | 2834 | 78.7 |
| Caregiver's age | | |
| ≥ 25 years | 2189 | 59.4 |
| Father's age | | |
| ≥ 30 years | 2621 | 71.7 |
| Father's education | | |
| ≥ 5 years | 2315 | 64.4 |
| Wealth index | | |
| Poor | 1236 | 33.2 |
| Middle | 1170 | 32.2 |
| Rich | 1228 | 34.6 |

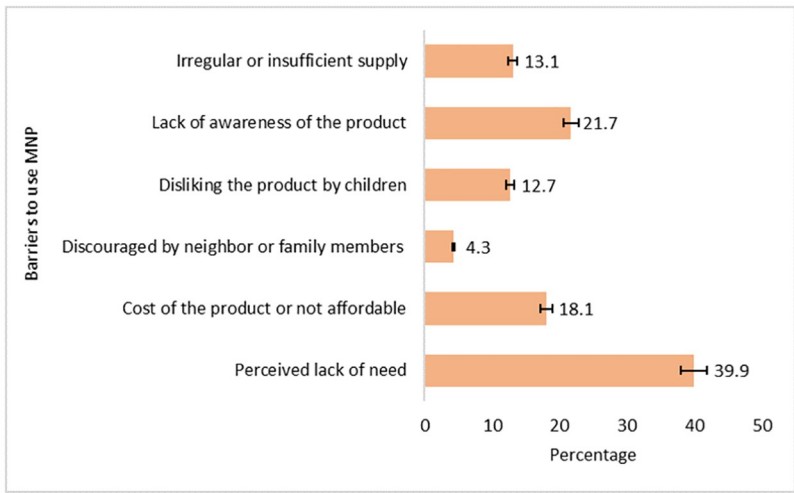

**Fig 2. Percentage of perceived barriers to the use of micronutrient powder reported by the caregivers of the children aged 6–59 months.**

unexpectedly, that caregivers who completed ≥ 5 years of schooling (aOR: 1.41, 95% CI: 1.13, 1.76) and caregivers of the children whose father completed ≥ 5 years of schooling (aOR: 1.38, 95% CI: 1.14, 1.67) were more likely to perceive lack of need for MNP for their children. However, caregivers of the households that had two or more children of 6–59 months were 20% less likely to perceive lack of need for MNP for their children compared to caregivers of the households that had one child.

Caregivers who perceived that cost of the product was a barrier to the use MNP, their such a perception was associated with the number of children of their household, their religious and educational status, their age, educational status of child's father, household wealth status and caregivers' IYCF practices. Caregivers of the households with two or more children aged 6–59 months (aOR: 1.51, 95% CI: 1.16, 1.96), caregivers of the Muslim households (aOR: 1.61, 95% CI: 1.02, 2.52) and caregivers of ≥ 25 years old (aOR: 1.35, 95% CI: 1.08, 1.69) were more likely to report that cost of MNP was a barrier to use. However, caregivers who completed ≥ 5 years of schooling (aOR: 0.50, 95% CI: 0.40, 0.63), caregivers from the middle (aOR: 0.59, 95% CI: 0.47, 0.76) and the rich (aOR: 0.26, 95% CI: 0.19, 0.35) households compared to poor households based on wealth status and caregivers who maintained optimal IYCF practices (aOR: 0.68, 95% CI: 0.53,0.88) were less likely to report the cost of the product as a barrier. Caregivers who received CHW's visit in the last 12 months were more likely (aOR 1.51, 95% CI: 1.06, 2.14) to report that they were discouraged by their neighbor or family members to the use MNP, compared to the caregivers who did not receive CHW's visit in the last 12 months.

Disliking the product by the children reported by the caregivers as a barrier was associated with child's age and sex, child's father age, CHW's visit in their households, their households' wealth status and their IYCF practices. Caregivers of the older children (aOR: 1.43, 95% CI: 1.11, 1.83), caregivers from the rich (aOR: 1.36, 95% CI: 1.05,1.77) and the middle (aOR: 1.40, 95% CI: 1.05,1.87) households in terms of wealth status and caregivers who received the visit of CHW in the last 12 months (aOR: 1.66, 95% CI: 1.31, 2.10) were more likely to report that their children did not like MNP, compared to their respective counterparts. On the other hand, the caregivers who were from the households of ≥ 5 members (aOR: 0.68, 95% CI: 0.54–0.86) and who maintained optimal IYCF practices (aOR: 0.75, 95% CI: 0.58, 0.97) were less likely to report that their children disliked MNP compared to the caregivers from the

**Table 2. Factors associated with the barriers to the use of micronutrient powder among the caregivers of the children aged 6–59 months.**

| Variables | Perceived lack of need | Cost of the product or not affordable | Discouraged by neighbor or family members | Disliking the product by children | Lack of awareness of the product | Irregular or insufficient supply |
|---|---|---|---|---|---|---|
| | aOR (95% CI) | aOR (95% CI) | aOR (95% CI) | aOR (95% CI) | aOR (95% CI) | aOR (95% CI) |
| Household size | | | | | | |
| < 5 members | 1 | 1 | 1 | 1 | 1 | 1 |
| ≥ 5 members | | | | 0.68** (0.54, 0.86) | 1.08 (0.88, 1.33) | |
| No. of children (6–59 months) in the households | | | | | | |
| One | 1 | 1 | 1 | 1 | 1 | 1 |
| Two or more | 0.80* (0.64, 0.99) | 1.51** (1.16, 1.96) | 1.41 (0.99, 2.00) | | 1.22 (0.95, 1.58) | |
| Sex of the children | | | | | | |
| Male | 1 | 1 | 1 | 1 | 1 | 1 |
| Female | 1.17 (1.00, 1.38) | | | 0.79* (0.63, 0.98) | | |
| Child's age | | | | | | |
| 6–23 months | 1 | 1 | 1 | 1 | 1 | 1 |
| 24–59 months | 1.69*** (1.43, 2.00) | 1.12 (0.90, 1.39) | | 1.43** (1.11, 1.83) | 0.23*** (0.19, 0.29) | 1.28* (1.02, 1.61) |
| Caregiver's religion | | | | | | |
| Hindu/Others | 1 | 1 | 1 | 1 | 1 | 1 |
| Muslim | 1.43* (1.05, 1.95) | 1.61* (1.02, 2.52) | | | 0.79 (0.60, 1.05) | |
| Caregiver's education | | | | | | |
| < 5 years | 1 | 1 | 1 | 1 | 1 | 1 |
| ≥ 5 years | 1.41** (1.13, 1.76) | 0.50*** (0.40, 0.63) | | | | |
| Caregiver's age | | | | | | |
| < 25 years | 1 | 1 | 1 | 1 | 1 | 1 |
| ≥ 25 years | | 1.35** (1.08, 1.69) | | | | |
| Father's age | | | | | | |
| < 30 years | 1 | 1 | 1 | 1 | 1 | 1 |
| ≥ 30 years | 1.18 (0.99, 1.40) | | | 0.77* (0.60, 0.98) | 0.81* (0.67, 0.99) | |
| Father's education | | | | | | |
| < 5 years | 1 | 1 | 1 | 1 | 1 | 1 |
| ≥ 5 years | 1.38** (1.14, 1.67) | 0.55*** (0.44, 0.69) | 1.34 (0.87, 2.06) | | 1.12 (0.92, 1.37) | |
| Wealth index | | | | | | |
| Poor | 1 | 1 | 1 | 1 | 1 | 1 |
| Middle | 1.45*** (1.19, 1.77) | 0.59*** (0.47, 0.76) | 0.91 (0.56, 1.47) | 1.36* (1.05, 1.77) | | |
| Rich | 1.81*** (1.46, 2.23) | 0.26*** (0.19, 0.35) | 1.20 (0.76, 1.89) | 1.40* (1.05, 1.87) | | |
| CHW's visit within last 12 months | | | | | | |
| No | 1 | 1 | 1 | 1 | 1 | 1 |
| Yes | 0.86 (0.73, 1.01) | 1.24 1.00, 1.55) | 1.51* (1.06, 2.14) | 1.66*** (1.31, 2.10) | 1.19 (0.98, 1.45) | 0.54*** (0.42, 0.68) |
| Optimal IYCF practices | | | | | | |
| No | 1 | 1 | 1 | 1 | 1 | 1 |

(*Continued*)

**Table 2.** (Continued)

| Variables | Perceived lack of need | Cost of the product or not affordable | Discouraged by neighbor or family members | Disliking the product by children | Lack of awareness of the product | Irregular or insufficient supply |
|---|---|---|---|---|---|---|
| | aOR (95% CI) | aOR (95% CI) | aOR (95% CI) | aOR (95% CI) | aOR (95% CI) | aOR (95% CI) |
| Yes | 1.32** (1.12, 1.57) | 0.68** (0.53, 0.88) | | 0.75* (0.58, 0.97) | 1.04 (0.85, 1.29) | |

*p value <0.05,

**p value <0.01,

***p value <0.001;

aOR = Adjusted Odds Ratio; CI = Confidence Interval.

households with <5 members and the caregivers who did not maintain optimal IYCF practices, respectively. Caregivers of the female children were 21% less likely (aOR: 0.79, 95% CI: 0.63, 0.98) to report that their children disliked MNP compared to the caregivers of the male children.

Lack of awareness of MNP among the caregivers was found to be associated with child age and child's father age. Caregivers of the older children were 77% less likely to report a lack of awareness as a barrier than the caregivers of younger children. The caregiver of the child, whose father's age was ≥ 30 years, was 19% less likely to report lack of awareness as a barrier compared to the caregiver of the child whose father's age was < 30 years.

Caregivers' perception of irregular or insufficient supply of MNP as a barrier was associated with child age and CHW's visit in their households. Caregivers of the older children were 28% more likely to report irregular or insufficient supply of MNP as a barrier compared to the caregivers of younger children. However, the caregivers who received the visit of CHW in the last 12 months were 46% less likely to report irregular or insufficient supply of MNP as a barrier compared to the caregivers who did not receive the visit of CHW in the last 12 months.

## Discussion

Evidence of the effectiveness of MNP programme targeting the younger children is limited. Several studies have identified a range of supply- and demand-side barriers to the use of MNP including lack of awareness of the products, lack of affordability and discouragement from the neighbors or other family members that corresponds with the findings of this study [10, 12, 14]. However, this paper further investigated underlying factors associated with the barriers to the use of MNP among the caregivers of the children and found several factors from supply-side and demand-side, associated with the barriers reported by the caregivers.

Caregivers' perceptions regarding the necessity of MNP for their children could rely on their previous experiences as the results indicate that perceived lack of need for MNP is less among the caregivers of more than one child. Study shows that number of children of the mothers does not influence their decision making in healthcare of the children [21], but it is very likely that previous experience of child rearing could create positive perception towards MNP if the previous child suffers from undernutrition. However, from our study findings we cannot assert that their previous experiences of child rearing make their perception positive towards MNP since we do not have data on nutritional status of the previous children.

Although unexpectedly we found that educated parents perceived more lack of need for MNP, it implies that formal education may not be sufficient to impart nutrition knowledge. A study indicates that educated mothers are very likely to be busy with their other prioritized works and that could result in their less concern about their children's nutrition [22]. Study

also illustrates that perception about health and nutrition can be improved through non-formal education whereas formal education can increase general knowledge [22]. So, our results underpin the necessity of social and behavior change communication to promote MNP among the caregivers of the children. It is evident that exposure to social and behavior change communication has substantial impact on complementary feeding practices of the children [23]. However, a study indicates that despite having recommended dietary diversity and meal frequency a child may not get adequate nutrients particularly iron, zinc and vitamin B6 because of bioavailability constraints of these problem nutrients in the foods [24]. Our findings show that caregivers who maintain optimal IYCF practices for their children perceive lack of need of MNP for their children. From caregivers' perspective, such a perception among them is usual as they may feel that they provide sufficient food to their children. It is also unexpected that they will understand that their children might lack of nutrients even if they feed recommended diets to their children, until they are communicated and informed.

CHW is one of the potential channels to communicate with the caregivers of the children at the community level [8], however, our findings show that the caregivers who received the visit of CHW are more likely to report that their children do not like MNP. It does not imply that the visit of the CHW is itself a barrier; rather due to the visit of the CHW more caregivers are receiving MNP from her but their children may dislike the product. Although results show that the caregiver who received the visit of the CHW are confronting discourage by the neighbor to use MNP, it is because the neighbor may not receive the visit from the CHW and thus they may be skeptical to the use of MNP received by the caregiver and de-motivate them. A study has illustrated those caregivers of the households that receive the visit from the CHW are likely to be informed about MNP and also to feed MNP to their children at least once, although there is a lack of evidence that such a visit can ensure compliance or effective use of MNP [18]. Therefore, the development of strategies such as counseling for the family members and community are important to address this barrier as suggested by the studies [14, 25]. However, different strategies should be taken into consideration at the initial stage of the programme implementation and it is recommended that implementing an MNP intervention in real-world contexts needs a comprehensive implementation research approach that would help implementers to address caregivers-level barriers from the very beginning of the implementation [26].

The CHW can play a vital role in counseling the family members and neighbors but in our study, CHW's visit has been found to have the potential to ensure the supply of MNP at the household level and it has a limited impact on motivating the family members and community to feed MNP to the children. So, during the visit of the CHWs to the households they should give more emphasis on describing the benefits of MNP and demonstration on how to feed MNP so that the caregivers can troubleshoot the problem related to the home fortification of diets with MNP. Although child's dislike of MNP is a crucial predictor of MNP intake [27], our study further shows that they are male children or older children who dislike MNP more as the caregivers reported, therefore visit of the CHW needs to be target-oriented, particularly at the households of the children who are reluctant to consume MNP.

The CHWs may intend to visit the households of the children who consume MNP because they target those households to increase the sale of MNP that could bring marginal profit for them. Thus, it could create a vicious cycle where the caregivers of the children who dislike MNP may claim that they do not feed their children because of the unavailability of MNP and the CHWs may pretext that they do not visit those households because the children of the households dislike MNP.

Based on the findings, the study suggests interventions that should consider the barriers from the perspectives of both the supply-side and demand-side. The interventions should

include a regular supply of MNP at an incentivized price, since insufficient supply and cost of the product have been found to be barriers to the use of MNP. On the other hand, dissemination of MNP promotional messages at the household and community level is warranted to build awareness of the need for MNP among the caregivers of the children.

## Strengths and limitations

This is the first study that has investigated perceived barriers to the use of MNP from the caregivers of the children of 6–59 months. This study not only identified the perceived barriers to the use of MNP but also explored the underlying factors. However, the study has some limitations. The study was conducted in some areas of Bangladesh and so the findings may not be generalizable for the entire country. We analyzed the sub-sample from the total estimated sample size which may reduce estimated power.

## Conclusion

This study indicates that rather than identifying the barriers to MNP use, researchers should investigate the root causes of those barriers. The programme intended to promoting MNP should carefully consider both the supply-side and demand-side factors that could underlie the barriers to the use of MNP.

## Supporting information

**S1 Table. Variables and scores used in constructing the infant and child feeding index.** (PDF)

## Acknowledgments

We thank all the respondents for participating in this study. We also thank the study team members who were involved in data collection and data management. icddr,b acknowledges with gratitude the commitment of CIFF to its research efforts. icddr,b is also grateful to the Governments of Bangladesh, Canada, Sweden and the UK for providing core/unrestricted support.

## Author Contributions

**Conceptualization:** Mahfuzur Rahman.

**Formal analysis:** Mahfuzur Rahman, Md. Tariqujjaman.

**Funding acquisition:** Haribondhu Sarma.

**Methodology:** Mahfuzur Rahman, Md. Tariqujjaman, Mustafa Mahfuz.

**Project administration:** Haribondhu Sarma.

**Supervision:** Tahmeed Ahmed, Haribondhu Sarma.

**Writing – original draft:** Mahfuzur Rahman.

**Writing – review & editing:** Mahfuzur Rahman, Md. Tariqujjaman, Mustafa Mahfuz, Tahmeed Ahmed, Haribondhu Sarma.

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
