## [Decision Letter · Decision Letter 0]

22 Jul 2021

PONE-D-21-19517

Caregiver perceived barriers to the use of micronutrient powder for children aged 6-59 months in Bangladesh

PLOS ONE

Dear Dr. Rahman,

Thank you for submitting your manuscript to PLOS ONE. After careful consideration, we feel that it has merit but does not fully meet PLOS ONE’s publication criteria as it currently stands. Therefore, we invite you to submit a revised version of the manuscript that addresses the points raised during the review process.

The paper has been reviewed by two independent reviewers. They have raised some concerns those need to be fixed before taking final decision. 

We look forward to receiving your revised manuscript.

Kind regards,

Enamul Kabir

Academic Editor

PLOS ONE

Journal Requirements:

2. Please ensure you have discussed the limitations of this study within the Discussion section, including any potential bias introduced during sampling and/or data collection.

Reviewers' comments:

Reviewer's Responses to Questions

**Comments to the Author**

1. Is the manuscript technically sound, and do the data support the conclusions?

Reviewer #1: Yes

Reviewer #2: Yes

2. Has the statistical analysis been performed appropriately and rigorously? 

Reviewer #1: N/A

Reviewer #2: Yes

3. Have the authors made all data underlying the findings in their manuscript fully available?

Reviewer #1: Yes

Reviewer #2: Yes

4. Is the manuscript presented in an intelligible fashion and written in standard English?

Reviewer #1: Yes

Reviewer #2: Yes

5. Review Comments to the Author

Reviewer #1: The authors report an ineresting survey of the underlying factors related to the barriers to use of MNP. The survey contains a large sample. I think the results are pretty sond and the data analysis is proper.

Reviewer #2: There is no denying that,MNP suppling is the major method for improving nutritional status and reducing anemia. In practice, however, PROTES barriers from the perspective of both supply and demand side.

Based on the demand side, this study analyzed the perception of the caregivers of the childrendemand side and socio-cultural factors.It can provide a basis for improvement in the follow-up health promotion programs.

It is suggested to further summarize and clarify that intervention plans on eliminating the obstacle factors.

6. PLOS authors have the option to publish the peer review history of their article (what does this mean?). If published, this will include your full peer review and any attached files.

Reviewer #1: No

Reviewer #2: No

---

## [Author Response · Author response to Decision Letter 0]

19 Aug 2021

We appreciate the very thoughtful reviews of the previous version of the manuscript. We have updated the text in response to the reviewers’ queries and feedback. A point-by-point response to each of the reviewers’ comments is included below. We believe these changes have substantially improved the manuscript. We hope you will find this revised manuscript appropriate for publication in PLOS ONE. Many thanks for your consideration.

Journal Requirements:

Response: Thank you for your valuable suggestions. We have checked the reference list and made corrections as per the formatting guidelines. There was an error in referencing ‘ Myatt M. IYCF assessment with small sample surveys: A proposal for a simplified and structured approach’ (#20 in reference list). Details on Infant and Child Feeding Index (ICFI) have been mentioned in ‘Guevarra E, Siling K, Chiwile F, Mutunga M, Senesie J, Beckley W, et al. IYCF assessment with small-sample surveys-A proposal for a simplified and structured approach. Field Exchange 47. 2014 Jul 19:60.’ Therefore, we have replaced the previous reference with this one in the revised manuscript.

 Response: Thanks. We have checked with the file formatting sample of PLOS ONE and made corrections in the title page. We have also made corrections in file naming as per PLOS ONE style templates.

2. Please ensure you have discussed the limitations of this study within the Discussion section, including any potential bias introduced during sampling and/or data collection.

 Response: Thank you for your observation. According to your suggestion, we have added a section on ‘Strengths and limitations’ within the Discussion section (Page 17, lines 355-361) in the revised manuscript. Two-stage random sampling technique was applied to select the caregivers of children from households and data collection was done by the trained data collectors, so we assume that no potential bias was introduced during sampling or data collection.

 Response: Thanks. We have checked the organization policy and revised the ‘Data Availability statement’ as per your suggestion as follows:

“Data generated from icddr,b’s research can be provided to interested researchers (Recipients) for secondary data analyses upon approval of a Data Licensing Application & Agreement by the icddr,b Data Centre Committee. Interested personnel is recommended to consult this with icddr,b IRB Coordinator Mr. M A Salam Khan (salamk@icddrb.org)”.

Response: Thank you for your observation. We have included caption for our Supporting Information file at the end of the manuscript and update the in-text citation accordingly.

Reviewers' comments:

5. Review Comments to the Author

Reviewer #1: The authors report an ineresting survey of the underlying factors related to the barriers to use of MNP. The survey contains a large sample. I think the results are pretty sond and the data analysis is proper.

Response: Thank you very much for your complements.

Reviewer #2: There is no denying that,MNP suppling is the major method for improving nutritional status and reducing anemia. In practice, however, PROTES barriers from the perspective of both supply and demand side.

Based on the demand side, this study analyzed the perception of the caregivers of the childrendemand side and socio-cultural factors.It can provide a basis for improvement in the follow-up health promotion programs.

It is suggested to further summarize and clarify that intervention plans on eliminating the obstacle factors.

Response: Thank you for your valuable comments and suggestions. According to your suggestions we have added some suggestive intervention plans based on our findings to eliminate the obstacle factors in discussion section (Pages 17, lines 348-353) of the revised manuscript.

---

## [Decision Letter · Decision Letter 1]

17 Nov 2021

Caregiver perceived barriers to the use of micronutrient powder for children aged 6-59 months in Bangladesh

PONE-D-21-19517R1

Dear Dr. Rahman,

We’re pleased to inform you that your manuscript has been judged scientifically suitable for publication and will be formally accepted for publication once it meets all outstanding technical requirements.

Kind regards,

Enamul Kabir

Academic Editor

PLOS ONE

Additional Editor Comments (optional):

Reviewers' comments:

Reviewer's Responses to Questions

**Comments to the Author**

1. If the authors have adequately addressed your comments raised in a previous round of review and you feel that this manuscript is now acceptable for publication, you may indicate that here to bypass the “Comments to the Author” section, enter your conflict of interest statement in the “Confidential to Editor” section, and submit your "Accept" recommendation.

Reviewer #1: All comments have been addressed

Reviewer #2: All comments have been addressed

2. Is the manuscript technically sound, and do the data support the conclusions?

Reviewer #1: Yes

Reviewer #2: Yes

3. Has the statistical analysis been performed appropriately and rigorously? 

Reviewer #1: Yes

Reviewer #2: Yes

4. Have the authors made all data underlying the findings in their manuscript fully available?

Reviewer #1: Yes

Reviewer #2: Yes

5. Is the manuscript presented in an intelligible fashion and written in standard English?

Reviewer #1: Yes

Reviewer #2: Yes

6. Review Comments to the Author

Reviewer #1: (No Response)

Reviewer #2: The influencing factors of MNP mentioned in this paper exist in some underdeveloped countries, and the research contents, problems found and suggested countermeasures proposed by the author will contribute to the improvement of children's nutrition.

7. PLOS authors have the option to publish the peer review history of their article (what does this mean?). If published, this will include your full peer review and any attached files.

Reviewer #1: No

Reviewer #2: No

---

## [Editor Report · Acceptance letter]

22 Nov 2021

PONE-D-21-19517R1 

Caregiver perceived barriers to the use of micronutrient powder for children aged 6-59 months in Bangladesh 

Dear Dr. Rahman:

I'm pleased to inform you that your manuscript has been deemed suitable for publication in PLOS ONE. Congratulations! Your manuscript is now with our production department. 

Kind regards, 

on behalf of

Dr. Enamul Kabir 

Academic Editor

PLOS ONE